# Back to the Future? Immunoglobulin Therapy for Myalgic Encephalomyelitis/Chronic Fatigue Syndrome

**DOI:** 10.3390/healthcare9111546

**Published:** 2021-11-12

**Authors:** Helen Brownlie, Nigel Speight

**Affiliations:** 1Independent Researcher and Former Local Government Officer, Social Policy and Research, Glasgow G2 4P, UK; hm.brownlie@virginmedia.com; 2Paediatrician and Independent Researcher, Durham DH1 1QN, UK

**Keywords:** immunoglobulin, myalgic encephalomyelitis, chronic fatigue syndrome, viral onset, cell-mediated immunity, post-acute sequelae of COVID-19, long-COVID

## Abstract

The findings of controlled trials on use of intravenous immunoglobulin G (IV IgG) to treat myalgic encephalomyelitis/chronic fatigue syndrome (ME/CFS) are generally viewed as representing mixed results. On detailed review, a clearer picture emerges, which suggests that the potential therapeutic value of this intervention has been underestimated. Our analysis is consistent with the propositions that: (1) IgG is highly effective for a proportion of patients with severe and well-characterised ME/CFS; (2) responders can be predicted with a high degree of accuracy based on markers of immune dysfunction. Rigorous steps were taken in the research trials to record adverse events, with transient symptom exacerbation commonly experienced in both intervention and placebo control groups, suggesting that this reflected the impact of participation on people with an illness characterised by post-exertional symptom exacerbation. Worsening of certain specific symptoms, notably headache, did occur more commonly with IgG and may have been concomitant to effective treatment, being associated with clinical improvement. The findings emerging from this review are supported by clinical observations relating to treatment of patients with severe and very severe ME/CFS, for whom intramuscular and subcutaneous administration provide alternative options. We conclude that: (1) there is a strong case for this area of research to be revived; (2) pending further research, clinicians would be justified in offering a course of IgG to selected ME/CFS patients at the more severe end of the spectrum. As the majority of trial participants had experienced an acute viral or viral-like onset, we further suggest that IgG treatment may be pertinent to the care of some patients who remain ill following infection with SARS-CoV-2 virus.

## 1. Introduction

Interest in researching immunoglobulin (IgG) therapy in patients with myalgic encephalomyelitis/chronic fatigue syndrome (ME/CFS) began in the mid-1980s, prompted by emerging evidence of immunoregulatory defects including disordered cell-mediated immunity (CMI) [1,2,3,4,5,6,7,8,9] and immunoglobulin subclass deficiencies [1,10,11,12]. A documented viral onset in some patients [2,13,14], the demonstration of enteroviral RNA in muscle tissue [15] and the presence of enteroviral antigen in serum [16] were viewed as supporting the hypothesis that ME/CFS may develop as a consequence of failed clearance of viral or other antigens [13,17] and strengthened the case for investigating an intervention predicated on the presence of immunologic dysfunction. High-dose intravenous (IV) IgG was known to ameliorate other disorders of immune regulation [1,18,19,20] and positive findings in a crossover study of intramuscular IgG therapy in patients with ‘chronic mononucleosis syndrome’ [21], in tandem with numerous individual case reports of beneficial outcome following IV IgG in patients with ME/CFS [18], provided further encouragement for this line of enquiry.

Four double-blind randomised controlled trials (RCTs) of IV IgG followed. Reports of effectiveness diverged [13,14,22,23,24], with findings collectively assessed as *“mixed”* [25] and *“inconclusive”* [26]. Following publication in 1997 of the last of these, [24] asserting an *“ineffective”* conclusion, interest in researching this treatment in patients with ME/CFS waned.

Recently, there have been renewed indications of research interest and the potential relevance of IgG in the treatment of patients with ME/CFS.

A detailed investigation of the effects of immunoglobulins on adrenergic receptors and immune function in ME/CFS was published in 2020 [27]. Additionally, in 2020, pilot work revealed that an autoimmune-associated small-fibre polyneuropathy (aaSFPN)—a condition known to respond to IV IgG—may occur in a substantial proportion of patients, leading the researcher to conclude: *“it may be important to identify ME/CFS patients who present with comorbid aaSFPN”* [28]. In March 2021, a paper reporting on immune-related pathogenesis revealed that several immunoglobulin genes are significantly increased in ME/CFS patients compared with controls [29]. On presenting at the 2020 International Association for Chronic Fatigue Syndrome/Myalgic Encephalomyelitis (IACFS/ME) Conference, the lead author expressed the clear opinion that findings indicate that therapeutic immunoglobulin studies are warranted [30]. Advice on the administration of IgG features in treatment recommendations produced by the US ME/CFS Clinician Coalition in February 2021 [31], while IgG is included in a summary of treatment approaches in an August 2021 paper from the Mayo Foundation for Medical Education and Research, setting out consensus recommendations for diagnosis and management [32].

Against this background, and with the recent lifting of a longstanding ban on use of UK-sourced blood plasma in the manufacture of immunoglobulins poised to ease global shortages of this product [33] and renewed interest in medium- to long-term immune dysfunctions following viral infection as post-acute sequelae of COVID-19 (PASC) [34,35], a fresh appraisal of this research, supported by published observations from clinical practice, is timely.

## 2. Nomenclature and Case Definitions

The research trials reviewed in this paper were published between 1990 and 1999 and refer to the disorder studied as ‘chronic fatigue syndrome’ or ‘the chronic fatigue syndrome’ (CFS). At that time, CFS was a recent entry to the medical lexicon, having been introduced in 1988 by the United States Centers for Disease Control (US CDC) with the stated intention of replacing the terms that were then current, i.e., ‘chronic Epstein–Barr virus syndrome’ or ‘chronic mononucleosis’, and variations [36]. ‘Myalgic encephalomyelitis’ (ME) was not utilised in the US but had been in common usage in the United Kingdom (UK), as was ‘post-viral fatigue syndrome’ (PVFS).

Hence, all of these identifiers are variously used to refer to the disorder studied in the respective papers cited from this period. The present paper will refer to ‘ME/CFS’, in keeping with current usage.

Amidst the range of possible descriptive terms, case definition for selection of patients remains the most crucial consideration with regard to assessing research: implications will be discussed.

## 3. The Immunoglobulin Trials

Four double-blind randomised placebo-controlled trials (RCTs) on the use of IV IgG to treat patients suffering from ME/CFS have been conducted. Results from the first two were published together in the American Journal of Medicine in November 1990 [13,14]. The respective authors had reached quite different conclusions. One study reported that immunoglobulin is effective in a *“significant number of patients*”, the other that IV IgG *“is unlikely to be of clinical benefit in CFS”.* This pattern was later repeated. In January 1997, an RCT reported a beneficial effect of IV IgG on adolescent patients [22]; later that year, a further, and to date final, trial reported that IV IgG *“is ineffective”* [24].

It is this apparently opaque and divergent picture that the present review seeks to illuminate. Scrutiny of recorded outcomes reveals that the findings of these trials were not mutually incompatible. To establish this, it is essential not to confine consideration to the most prominently reported conclusions, but to take into account features of the respective studies and their specific findings.

For simplicity, we shall refer to these four trials as Studies 1 [13], 2 [14], 3 [22] and 4 [24]. Table 1 and Table 2 illustrate trial characteristics and participants, respectively.

The measures employed to record the nature and severity of symptoms experienced and their impact on daily life varied from study to study, comprising a mix of established protocols [40,41,42,43,44,45,46,47,48,49], modifications of established protocols [50], and rating methods devised by the authors. These are listed in Appendix A.

All trials obtained measures of cell-mediated immunity and, with the exception of Study 2, these were considered at baseline and at follow-up stage; Study 2 focused on humoral immunity (IgG subclass levels) when assessing outcomes. The tests of cell-mediated immunity performed in the respective studies are listed at Appendix B.

All trials were double blinded; however, details of exactly how the blinding of investigators was conducted are not provided. Studies 1 and 3 record that the follow-up assessment was conducted by a physician who was unaware of the treatment regimen.

All four trials took rigorous steps to record and respond to any adverse experiences. This aspect is summarised and discussed at Section 5.1 below.

### 3.1. Study 1 (Australia) Lloyd et al., 1990

This trial involved 49 participants aged 16–63, randomised to receive a total of three infusions of IV IgG at 2 g per kilogram of bodyweight (*n* = 23) or placebo (*n* = 26), administered at monthly intervals [13].

#### 3.1.1. Cell-Mediated Immunity at Baseline

Two-thirds of participants (33) had reduced cutaneous delayed-type hypersensitivity (DTH) responses, while 40 (82%) had abnormal CMI in this form and/or as evidenced by T-cell lymphopenia. CD8 cell count was below the normal reference range for the authors’ laboratory in 18 participants (37%) and CD4 in nine (18%), while the absolute T-cell count was below the reference range in 21 (43%).

#### 3.1.2. Summary of Reported Findings

Three months after the final infusion, 10 immunoglobulin recipients (43%) and three placebo recipients (12%) were designated as ‘responders’ on blinded evaluation by physician following participant interviews where specific details of employment, social and leisure activities were obtained. The higher incidence of ‘responders’ in the IV IgG group was statistically significant.

On this basis, the authors conclude: *“The results of this study demonstrate that a significant proportion of patients (43%) with well-characterized, severe, and long standing CFS responded to high-dose intravenous immunoglobulin therapy.”*

It is notable that this categorisation of participants as ‘responders’ or ‘non-responders’ was not a matter of fine judgment. All ‘responders’ had improved considerably: “*This response was characterized by recommencement of employment, leisure, and social activities, as well as by a significant reduction in physical and psychologic morbidity and by an improvement in cell-mediated immunity*.” (see Box 1 below).

Box 1Employment, Leisure and Social Activities—three months after final infusion, Lloyd et al., 1990 [13]; IgG ‘responders’ *n* = 10; placebo ‘responders’ *n* = 3. 
**Employment and Household Duties:**
Six had resumed pre-morbid employment status in full-time occupation or housework; three had been working on a part-time basis and were now able to work full-time; one had been unemployed before contracting ME/CFS and remained so, but was now able to resume his leisure activity of bushwalking. *One had been in full-time work before trial commencement and remained so; two moved from part-time to full-time work.*
**Leisure Activities:**
None had been involved in any leisure activities prior to the trial; all but one were now participating in leisure activities including tennis, surfing and gardening; the person who did not resume any leisure activities was now working full-time as a labourer.
*Two resumed tennis and swimming, respectively; the third person, described as a ‘housewife’, did not participate in leisure activities before or after the trial.*

**Social Activities:**
All had increased participation in social activities; seven were now involved in social events at least once a week, three who had previously managed no social activity were now participating in social events less often than once a week.
*All had increased participation, one to at least once a week.*


In contrast, the ‘non-responders’ had experienced little or no change, a pattern that emerged consistently across the range of outcome measures: “*The distribution of the change in the self-report measures of physical and psychologic morbidity fell in a dichotomous (i.e., almost “all or nothing”) pattern, consistent with the categorized “response” or “no response” assessment from the physician’s interviews.”*

Every ‘responder’—whether in the IgG or placebo group—who underwent immunologic follow-up was found to have experienced a significant improvement in cell-mediated immunity (CMI). These twelve patients (one placebo ‘responder’ was unable to undergo immunological testing at follow up) recorded a significantly greater improvement in CD4 (helper T) cell count when compared with the 36 participants designated as ‘non-responders’, with the group average up 37% versus down 3% (*p* < 0.01). Reduced DTH responses, present in eight at entry, had resolved to normal values for all but one.

The authors observe: “*The association between recovery from CFS and resolution of abnormalities in cell-mediated immunity is strengthened by the demonstration of immunologic improvement in both the placebo responders tested at follow-up*”.

Among the 23 IgG recipients, the percentage change in score on a modified Quality of Life visual analogue scale (see Appendix A) was positively correlated with improvement in CD4 count (*r* = 0.4, *p* < 0.05) and with improved DTH (*r* = 0.3, *p* = 0.08).

Most of those who benefitted from IgG had improved promptly: “*in 8 of the 10 IgG ‘responders’, improvement in symptoms and function was noted within 3 weeks of the first infusion and tended to increase incrementally after subsequent infusions*”.

Further therapy was able to reverse any subsequent decline. Clinical follow-up identified that, by one year after the final infusion, eight of the ten IgG ‘responders’ “*had substantial return of symptoms and disability*”. These patients received further IgG therapy *“that was associated with an identical remission of symptoms in improvement in function in each case”.*

#### 3.1.3. Discussion of Study 1

PARTICIPANT CHARACTERISTICS The assertion that participants were patients whose illness was *“well-characterized, severe and longstanding”* is borne out by the recruitment criteria employed [51]. All were required to experience *“marked exercise-aggravated muscle fatigue, with abnormally prolonged recovery time”,* ensuring that a core defining feature of ME/CFS was present.

OUTCOMES Described as *“almost ‘all or nothing’”*, the distribution of outcomes *“prevented significant differences in physical, psychologic, or immune measures from being demonstrated between the immunoglobulin and placebo recipients when analyzed as a group”*. Therefore, confining analysis to a comparison of intervention and control group means on conclusion of the trial—a widely accepted method of judging efficacy—would have concealed the significantly greater likelihood of the patient experiencing clear improvement if treated with IgG.

PREDICTORS OF OUTCOME Predictors of improvement were found to lie in immunologic measures recorded at entry, particularly a lowed CD4 lymphocyte count: *“supporting the concept that cellular immunity is important in both the pathogenesis and response to treatment in patients with CFS.”* The analysis conducted implies that ME/CFS patients who have scope to benefit from IgG may be predicted in advance with a high degree of accuracy. A separate analysis based on age, sex and disease duration did not significantly predict response.

### 3.2. Study 2 (USA) Peterson et al., 1990

This trial involved 30 participants aged up to 74, with the IV IgG group (*n* = 15) receiving half the dose used in Study 1, at 1 g per kilogram of bodyweight. A total of six infusions were administered, at 30 day intervals [14].

#### 3.2.1. Cell-Mediated Immunity at Baseline

Five participants (18%) had reduced numbers of T3 (pan T) cells; five had reduced numbers of CD4 cells. CD8 cells were low in three participants (11%). This was the only study to assess B cells: none had a reduced number of B lymphocytes; however, an abnormally high number was found in two patients (7%).

IgG subclass levels—reflecting humoral immunity—were also assessed in this trial. Low levels of IgG1 were found in 12 participants (43%) and low levels of IgG3 in 18 (64%).

#### 3.2.2. Summary of Reported Findings

The final outcome point involved ratings of the period between penultimate and final infusions. Comparison of intervention and control group means on various measures indicated that the only statistically significant difference between the IgG and placebo control groups over the course of the trial had been *“a slight relative improvement in social functioning of the placebo group”*. However, significant changes had occurred for the respective groups. An improvement in health perceptions was found in the IgG group, where mean score on the relevant scale of the Medical Outcome Study short-form (MOS SF) [42,43] had risen from 8.5 to 20.5 (on a scale of 1–100), an increase of 12 points. Group mean score on physical function, also measured by MOS SF, had significantly deteriorated in the placebo group.

Regarding impact on IgG levels, when assessed within 48 h of the final infusion, all participants recorded IgG 1 levels within the normal range, the deficiency recorded in 12 patients at baseline having resolved. On the other hand, IgG3 had normalised in only six of the 18 patients where it had previously been abnormal. The authors note that this is *“consistent with the shorter half-life of IgG3 (7 days)* versus *IgG1 (22 days)”*, continuing: “*If a deficiency of IgG3 is of pathogenic importance, this could provide a pharmacodynamic explanation for treatment failure*.”

Cell-mediated immunity was not assessed after baseline.

The authors observe that findings *“do not, of course, rule out a potential benefit of other immunoglobulin preparations, dosage regimens, or longer courses of therapy”.* Nonetheless, their prominent conclusion was that *“IV IgG is unlikely to be of clinical benefit in CFS.”*

#### 3.2.3. Discussion of Study 2

SAMPLE SIZE Based on the hypothesis of symptomatic benefit of IgG in 67% and placebo in 25%, the authors had calculated that a trial size of 30 would have sufficient power to detect a statistically significant difference. However, the marked improvements observed in Study 1 had occurred in a lower proportion: 43% of IgG and 12% of placebo recipients. An analysis involving a trial size of 28 (one participant was lost from each group) therefore lacked the power to detect change occurring at the level observed in Study 1.

RESULTS It is remarkable that the significant improvement in health perceptions unique to the IV IgG group did not temper the negative conclusion presented. The authors describe all recorded changes as of a *“small magnitude”* and therefore of questionable clinical importance. It is far from apparent that the scale of this change—from a mean of 8.5 to 20.5 i.e., +141% —was of a *“small magnitude”.* Furthermore, the standard deviation was substantial (25.0), implying that some of the IgG group recorded an improved level of health perception following treatment that was far in excess of the group mean.

There was no comparable analysis to that carried out in Study 1, which identified a distinct pattern of change—i.e., participants improved greatly or scarcely at all—a change that was masked when group means were compared.

PARTICIPANT CHARACTERISTICS Recruitment involved CDC 1988 criteria [36], where post-exertional deterioration is optional. However, according to symptom reports all but one participant did experience prolonged fatigue post-exertion.

Participants appear to have had less severe ME/CFS as mean scores on the MOS SF Physical Function Subscale [42,43] at baseline were over 60 for both groups. In contrast, a paper reporting the function of patients in Prof Peterson’s clinic from around this time indicates that mean scores on this scale were in the region of 16 [52].

PREDICTORS OF OUTCOME Had patients whose health perceptions improved also experienced improvements in CMI, as identified in Study 1? As CMI was not assessed after baseline, we simply do not know.

IgG subclass levels (humoral immunity) were recorded post-treatment (see Section 3.2.2), but were not correlated with other outcome measures.

### 3.3. Study 3 (Australia) Rowe 1997

This trial involved the administration of IV IgG (*n* = 35) or placebo (*n* = 35) in adolescents aged 11–18 [22]. A subsequent paper [23] included further analysis of trial data.

#### 3.3.1. Cell-Mediated Immunity at Baseline

On cutaneous delayed-type hypersensitivity (DTH) testing, a complete absence of CMI response (anergy) was found in 21% of patients tested at baseline, while a further 31% evidenced a reduced response (hypoergy). (This information relates to 63 of the 70 participants in this trial.) In contrast, normative data for healthy children indicate that only 5% are hypoergic and none are anergic [53].

#### 3.3.2. Summary of Reported Findings

Outcomes on a range of measures were positive for both intervention and placebo control groups, but with significantly greater improvement occurring in the IgG group when assessed six months post the final infusion.

The primary outcome measure involved an overall functional score that was based on proportion of: school attendance; school work attempted; normal social activity; and of normal physical activity attempted. The young people were to compare themselves with premorbid levels and a mean of the four ratings was calculated for each participant. On a scale of 0–100, where 100 represents premorbid level of activity, the mean functional improvement from baseline was significantly greater for the IgG group (from 23.9 to 64.1, up 40.2 points) than the placebo control group (from 25.9 to 52.1, up 26.2 points). The proportion who had returned to full function was significantly greater also: 25% of IV IgG recipients, as compared to 12% in the placebo control group.

It is notable that significant differences between IgG and placebo groups emerged at the six month follow-up stage. Follow-up at three months after the final infusion found improvements of a lesser degree, which were not statistically significant. For example, the incidence of a *“clinically significant improvement”—*set at a functional improvement of at least 25% over baseline—was 52% among IgG recipients at three months and 72.2% at six months (see Table 3).

A subsequent paper [23] presented further analysis of the data, investigating the role of CMI in the form of DTH in determining outcomes. It was found that young ME/CFS patients with an appropriate CMI response had tended to experience improvement, whether treated with IgG or not, while those with impaired response fared better only if treated, as those with poor CMI treated with IgG recorded an average functioning score at six-month follow-up of 65.4, as compared to 39.3 for their counterparts in the placebo control group.

#### 3.3.3. Discussion of Study 3

PARTICIPANT CHARACTERISTICS This remains one of very few RCTs investigating the efficacy of a pharmacological intervention specifically in young people with ME/CFS. The young patients were reported to be *“by selection at the more severe end of the spectrum”* [23]. They were required to meet 1994 CDC Criteria [39] and to be experiencing *“chronic and persisting or relapsing fatigue of a generalized nature, exacerbated by minor exercise”.* The latter condition represents a strengthening of these criteria, in which *“postexertional malaise lasting more than 24 h”* is merely optional.

PREDICTORS OF OUTCOME In addition to consideration of outcomes in relation to DTH (see Section 3.3.2 above), analysis was conducted to identify any associations between side effects and outcome. *“Severity of headache”* and *“duration of nausea”* proved to be sensitive for predicting clinical improvement. These were not specific to clinical improvement, as all but one of those who improved only minimally also experienced them, suggesting that experience of severe headache and nausea of more than three days duration following infusion may have been a necessary but not a sufficient condition of clinically effective action of IgG on the patient.

### 3.4. Study 4: (Australia) Vollmer-Conna et al., 1997

Several authors of Study 1, together with others, went on to conduct this further trial *“To determine whether the reported therapeutic benefit of intravenous immunoglobulin in patients with chronic fatigue syndrome is dose dependent”* [24]. Participants received either placebo or IgG infusion at one of three doses: 2 g per kg, as used in Study 1—described as ‘high’ (*n* = 23); 1 g per kg, as used in Study 2—described as ‘medium’ (*n* = 28); and 0.5 g per kg—described as ‘low’ (*n* = 22).

#### 3.4.1. Cell-Mediated Immunity at Baseline

At entry, abnormal CMI in the form of reduced DTH skin responses was evident in 47 (48%) with 16 of these patients (16%) showing anergy. However, the proportion evidencing reduced DTH reponses varied greatly between the study groups, ranging from 32% in the medium-dose IgG group to 64% in the high-dose group.

#### 3.4.2. Summary of Reported Findings

The published paper presents a stark conclusion, placed prominently in the title: *“Intravenous Immunoglobulin Is Ineffective in the Treatment of Patients with Chronic Fatigue Syndrome”* [24]. Far from showing that participants did not experience improvement, it was reported that three months after the final infusion *a**ll* groups had improved significantly, regardless of whether infused with IgG or placebo and regardless of the IgG dose administered. For example, the increase in Karnofsky performance scores [48] pre- and post -treatment is reported as highly significant in all groups, but with no significant difference *between* the groups.

Furthermore, all groups had experienced improvement in measures of cell-mediated immunity over the same time period. Immunological findings are presented as group means for CD4 and CD8 cell counts at baseline, immediately before the final infusion and at follow-up three months later. It is reported that *“A significant, linear increase in the absolute numbers of T suppressor/cytotoxic (CD8) cells was demonstrated across the three measurement occasions (F = 17.8, p < 0.0001).”* T inducer (CD4) cells, on the other hand, had flatlined (high-dose IgG group) or reduced (all other groups) over the course of the trial, then increased during the following three months—though the mean count for the medium-dose IgG group remained below baseline level.

#### 3.4.3. Discussion of Study 4

In view of the subsequent influence of its ‘negative’ conclusion, we present a more extensive commentary on this particular paper.

PARTICIPANT CHARACTERISTICS: The authors advise that participants were “*probably, more heterogeneous than that enrolled in our earlier study*”, i.e., Study 1 [13]. This assertion appears to be justified. Recruitment criteria [37,38] lacked an indicator of post-exertional deterioration. Being severely affected was not the norm: all four groups were undertaking a mean of five hours of non-sedentary activity per day at baseline, with some participants managing in excess of nine hours of non-sedentary activity.

RESULTS: Having observed that *“improvements occurred irrespective of whether patients received immunoglobulin or placebo infusions”,* the authors conclude: *“This outcome, although consistent with the results obtained by Peterson et al., does not support our own previous findings of significant clinical and immunologic improvement with high (2 g/kg) dose immunoglobulin”.* Yet, the group given 2 g/kg dose of IgG had improved significantly in respect of both clinical and immunological measures in this trial. What was different here from their previous trial (Study 1) was the occurrence of significant improvement in the placebo group also.

RATIONALE FOR IMPROVEMENT IN PLACEBO GROUP: The authors speculate that significant improvements in functioning may simply reflect the natural course of the illness. This is unconvincing: a 2005 systematic review of 14 studies found that fewer than half of patients had improved [54]. A more likely explanation may lie in the operation of the ‘placebo’ preparation. This contained albumin, which exerts high oncotic pressure, acting to improve brain plasma volume. (Studies 2 and 3 also involved a placebo containing albumin, while in Study 1 this is unclear.)

Regarding blinding in this study, the placebo solution was identical in appearance to the IV IgG solution (Intragram™). Both solutions were delivered to the pharmacies of participating institutions in 500-mL bottles.

PATTERN OF CHANGE: Study 1 had likewise found no significant difference between intervention and control groups on conclusion of the trial, when assessed as a whole, but had observed a dichotomous distribution of outcomes in both groups. Patients improved greatly—being referred to as ‘recovered’—or scarcely at all. It is notable that the investigators did not interrogate the data from the present study for the presence of a ‘response’/‘non-response’ pattern.

The rationale provided for not doing so is that the method used—being based on physician’s assessment of substantial improvement in functional capacity and symptomatology—*“is potentially too subjective”*.

We find this unconvincing. This perspective on the primary outcome measure used in Study 1 is irrelevant: Study 4 did not employ this outcome measure. The data emerging on the outcome measures that *were* employed in Study 4 could have been interrogated to establish if a similar pattern of strong response vs. little or no response pertained.

With regard to this retrospective criticism of their own method, the issue is not whether or not a physician assessment could possibly be *“too subjective”,* but whether or not this could rightly be said of the method employed in Study 1. We are of the view that this charge would be misplaced and that the assessments of outcomes recorded in Study 1 were robust, for three reasons. Firstly, the patient had to agree with the physician’s assessment that they were a ‘responder’. Secondly, ‘response’ vs. ‘non-response’ designation was arrived at with reference to specific details of real-life activities. Thirdly, physician ratings were corroborated by other outcome measures.)

BETWEEN GROUP DIFFERENCES The data presented do not appear to bear out the narrative conclusion concerning the lack of any discernable difference between the groups, with the ‘medium’-dose IgG group recording a median incremental change on the Karnofsky Scale [48] four times the magnitude occurring in the ‘low’-dose group (10 points, as compared to 2.5 points).

PREDICTORS OF OUTCOME Study 1 had found resolution of abnormal CMI to be strongly predictive of response. There was no attempt to assess CMI against improvements in function on a patient by patient basis for degree of association in the present Study. This is remarkable as, in their discussion, the authors observe: *“If one could identify a cohort of genuine postinfective cases with CFS as well as immunological abnormalities, then it may be appropriate to reevaluate intravenous immunoglobulin therapy in that specific group only”.*

### 3.5. Summary of Immunoglobulin Trial Findings

STUDY 1 [13] found that “*A significant proportion of patients (43%) with well-characterized, severe, and long standing CFS responded to high dose intravenous immunoglobulin therapy**”*. Likely responders could be identified in measures of cell-mediated immunity (CMI) at baseline. Change was either dramatic—being referred to as *“recovery”—*or negligible. Participants who subsequently relapsed were given further IgG therapy, *“associated with an identical remission of symptoms in improvement in function in each case”.*

Despite differing ‘headlines’ emerging from the following studies, none recorded outcomes that are demonstrably inconsistent with these clear findings from Study 1.

STUDY 2 [14] was small and lacked the power to identify the incidence of change observed in Study 1. The significant improvement in health perceptions, unique to the IgG group, is not discussed or referred to by the authors when reaching conclusions.

STUDY 3 [22,23] found a significantly greater improvement among adolescent ME/CFS patients treated with IV IgG in comparison with placebo. Furthermore, this improvement was of a clinically significant magnitude. It also found that young patients with normal CMI tended to improve over time, regardless of treatment, while those with subnormal CMI were significantly more likely to improve if treated with IgG.

STUDY 4 [24] involved the least well-characterised group of patients and the least severely affected. All groups, including placebo, improved significantly and all had experienced improvements in CMI. There was no attempt to ascertain if resolution of abnormal CMI was associated with individual participant improvement. Nor did the authors interrogate the data for evidence of the ‘response’/‘non-response’ dichotomy which had emerged in Study 1, and which would have remained masked had analysis been confined to comparison of IgG and placebo group mean scores, as in Study 4.

PARTICIPANT CHARACTERISTICS It was a pre-condition of recruitment to Studies 1 and 3 that the patient’s clinical picture include post-exertional deterioration. While not a recruitment criterion in Study 2, all but one participant did report experiencing *“prolonged post-exertional fatigue”.* Study 4 was the exception in this regard.

PLACEBO GROUPS In so far as some placebo group participants had improved, this was found to be associated with resolution of, or improvement in, abnormal cell-mediated immunity (investigated in Studies 1 and 3). It is further plausible that the ‘placebo’ infusion had intrinsic therapeutic action. Albumin, used in the placebo infusion in at least three of the four trials (Studies 2, 3 and 4), is known to exert high oncotic pressure leading to improved perfusion of vital organs, including the brain.

## 4. Clinical Experience

Following the publication of Study 4 [24] in November 1997 no further RCTs of immunoglobulin treatment for ME/CFS have been conducted. In contrast, reports in the peer-reviewed medical literature document clinical experience that is predominantly positive.

Published reports concerning use of IgG in clinical practice include a 2003 article from the UK ‘Successful Intravenous Immunoglobulin Therapy in 3 Cases of Parvovirus B19–Associated Chronic Fatigue Syndrome’ [55] and a 2013 paper documenting outcomes in respect of treatment schedules followed by patients attending the CFS Unit of the Aviano National Cancer Institute, Italy, which found that antiviral and immunological therapies (jointly reported) were the most successful, with 75% of those treated responding [56]. The authors of the 2013 paper conclude: *“our results show that a significant number of patients treated with antiviral/immunoglobulin approaches have a long positive disease free survival in comparison with other patients treated with other approaches”* and describe these treatments as *“clearly superior”*.

Conference presentations provide another source of positive reinforcement for the potential benefits of IgG. In the US, Dr James Oleske—then chair of the Chronic Fatigue Syndrome Advisory Committee (CFSAC) to Health and Human Services (HHS)—made a presentation to CFSAC in 2009 reporting on successful IgG treatment of young patients [57]. More recently, Ryan Wheelan of Simmaron Research presented to the 2020 IACFS/ME conference data from a patient who had neuropathic symptoms in the presence of an autoimmune-associated small-fibre polyneuropathy (aaSFPN) [28]. Following two IV IgG infusions, *“The patient experienced a dramatic reduction in levels of all four of the relevant autoantibodies and favorable symptom reduction*.”

### 4.1. Brief Case Reports (NS)

One of the authors, (NS), has significant experience of using immunoglobulin in selected cases of severe ME/CFS, including three cases described elsewhere in this issue [58]. Thumbnail sketches of their IgG treatment together with the experiences of two other young patients are presented below. 

These cases occurred over a sixteen-year period between 1990 and 2006. Most had clinical evidence of a viral onset. All but one received IgG treatment by intra-muscular (IM) administration, being too unwell to attend hospital.

Interest in the therapeutic use of immunoglobulin stemmed from a case where the family doctor, unbeknown to paediatrician NS, started his patient on IM immunoglobulin. NS observed that the patient made a rapid improvement over the next few months and managed to return to full time schooling after a total loss of two years; thereafter he continued to improve and was able to participate in sports.

Following this experience, NS decided on a policy of offering a trial of immunoglobulin to each of his severely affected cases, supported by the evidence in the Lloyd et al. paper [13] (Study 1, Section 3.1 above; the positive adolescent study by Rowe [22,23]—Study 3, Section 3.3—not yet having been published) and awareness of the research base on immune system defects.

Prescription of immunoglobulin was based on the clear clinical picture of ME/CFS. All would have fitted any of the main diagnostic definitions now in use. and in the course of long-term clinical monitoring and follow-up none transpired to have been misdiagnosed. 

#### 4.1.1. Case 1

This 13-year-old girl was bedridden and needed tube feeding over a period of 18 months. Testing of blood samples identified that antibody titres to Coxsackie virus were raised; this remained the case for one year. She was given monthly injections of IM immunoglobulin and made a slow but steady recovery over a two year period. In her 20s, she was fully recovered and holding down a full-time job.

#### 4.1.2. Case 2

This 13-year-old girl had a sudden onset of very severe ME and lay in a darkened room in severe pain for over 12 months. She was then given IM immunoglobulin for a year and made a total recovery over two years.

#### 4.1.3. Case 3

This adolescent boy had moderately severe ME for over a year. He was well enough to come to hospital for monthly IV immunoglobulin. His condition improved slowly over two years, and he eventually made a full recovery, being able to undertake strenuous athletic pursuits.

#### 4.1.4. Case 4

This 13-year-old girl was the most severe case of this series, being virtually paralysed with shallow respirations for several months. She received IM immunoglobulin but there was no improvement over the first nine months. She then began to improve steadily, and made a total recovery over two years. Ten years later, she remained completely well.

#### 4.1.5. Case 5

NS encountered this 19-year-old young woman through meeting her parents at an ME conference in Norway. She had been lying in a darkened room for several years and was not improving. On his suggestion, her family doctor gave her a trial of immunoglobulin, following which she improved rapidly. Within nine months, she had a functional level of 90% and was cycling and campaigning for her chosen party in elections.

#### 4.1.6. Reflection

While this is a series of documented clinical histories that is uncontrolled, it is worth stressing that: (a) all the cases were severe or very severe; (b) all made near total recoveries; and (c) there were no treatment failures, only instances of treatment success. Treatment continued until response was achieved, in contrast to the research trials which provided shorter-term treatment. Additionally notable in a clinical context is the fact that patients in Study 1 who relapsed after the trial ceased were given further IgG treatment, which effected an identical remission in each case (as reported at Section 3.1.2 above).

## 5. Informed Consent: Awareness of Any Potential for Adverse Events

It is of course vital that both physician and patient are aware of potential for adverse events before commencing any treatment or therapy. Informed consent involves the patient being provided with the information required to assess any potential adverse events against the health-related impact on quality of life of their ongoing illness and to contrast this with any alternative therapeutic options.

Quality of life with this illness is low if left untreated [59,60,61,62,63,64,65,66,67,68].

Information given to patients receiving IgG on the National Health Service (NHS) in the UK advises that “*some patients may experience reactions during treatment such as fevers, shivering, skin rashes, wheezing and headaches*” [69]. According to *‘Managing patients with side effects and adverse events to immunoglobulin therapy’*—a 2016 paper from the Expert Review on Clinical Pharmacology [70]—*“Most of the adverse effects associated with immunoglobulin therapy are mild, transient and self-limiting”* while, less commonly, serious side effects can occur. The latter may be inherent in the mode of operation as key issues identified include: “*The antigenicity of the IgG, large molecular weight of the IgG aggregates, and complement activation or direct release of cytokines from mononuclear cells may be underlying mechanisms of adverse reactions in immunoglobulin therapy*.”

The findings of the IgG trials involving ME/CFS patients reflect these assessments, with the additional dimension that the effort required to participate could cause an exacerbation of ME/CFS.

### 5.1. Summary of Adverse Events Reported in IgG Trials Involving Patients with ME/CFS

Adverse events commonly took the form of a transient exacerbation of ME/CFS symptoms and for the most part were equally likely to occur in the IgG and placebo control groups, suggesting that they were attributable to the effort of participation in the research.

Some adverse experiences, notably severe headache, were more commonly reported among participants receiving IgG and as such may be treatment related. Furthermore, this may have been intrinsic to effective operation of IgG as they were found to be correlated with a positive outcome in Study 3 (discussed at Section 3.3.3 above).

A small number of unspecified *“major adverse experiences”* were reported in Study 2, with the same incidence in IgG and placebo control groups (three of 15 patients in each). It is not clear that any of these were specifically related to the administration of IgG.

Study 3 took steps to obtain feedback from the young patients regarding what might be improved in this regard, reporting: *“Seventy-five per cent thought they received positive benefit from participating in the trial but 30% thought that certain aspects could have been managed better. This included management of side effects, the option to stay overnight rather than treatment in the Day Medical Ward as many had to travel long distances, and prior information about the likely severity of symptoms after the infusions”* [23].

### 5.2. Discussion and Implications

As with any therapy, steps can and should be taken to minimise the potential for adverse impact of IgG treatment on a patient with ME/CFS, such as those suggested by the young patients in Study 3 (see Section 5.1 above).

Intramuscular and subcutaneous administration present alternatives to IV, avoiding the necessity of travel to hospital. These may be preferable options for the most severely affected patients.

## 6. Healthcare Systems and Immunoglobulin for ME/CFS

Immunoglobulin therapy remains unendorsed in healthcare guidance for ME/CFS in the USA [71,72], Australia [73], Canada [74] and the UK [75]. Negative recommendations tend to rest on two elements. Firstly, the assessed results of the four trials, historically summarised as *“mixed”* [25], *“inconclusive”* [26,76], and/or *“insufficient”* [76]. Secondly, the reported incidence of adverse events. While a full analysis is beyond the scope of this paper, both elements would appear to have been subject to a degree of misinterpretation and selective reporting, as exemplified in the justification provided for the UK’s *“not recommended”* decision (described at Section 6.1.1. below). We have covered both aspects in this paper (in Section 3 and Section 5, respectively) drawing on the original trial reports [13,14,22,23,24] and hold that a strong case for IgG therapy in selected cases of severe ME/CFS emerges.

### 6.1. Accessing Immunoglobulin Therapy

#### 6.1.1. United Kingdom

Faced with increasing therapeutic use of IgG and reduced supply, guidelines limiting use on the NHS were introduced in 2008, with ‘chronic fatigue syndrome’ listed under *“indications for which IVIG is not recommended”* [77] and this remains the case in the current version, published in 2011 [78]. As none of the research studies on IgG for ME/CFS were among the 248 publications referenced, the evidence used to reach this decision was opaque, emerging only in reply to rapid responses challenging this assessment following publication of a summary in the British Medical Journal [79].

The reply advises: *“The principal study reviewed (a randomized, double-blind, placebo controlled trial) showed no significant benefit of intravenous immunoglobulin”* [80], referencing Study 4 [24]: this trial had found significant improvements in all groups, including placebo. There is no acknowledgement of the clearly positive findings for IgG over placebo reported in Studies 1 [13] and 3 [22,23]. It is further asserted that findings raise the potential for adverse reactions. It is not noted that worsened symptoms had commonly been reported to occur as just commonly in the placebo control group as well as among those receiving IgG in this trial. Rather, the reference is specifically to *“potential and sometimes fatal risks associated with treatment with a blood product”*. We are not aware of any other disorder where the case for therapeutic use of IgG is inhibited on the grounds of the potential risk of blood products.

Rapid responses [81,82] highlighted a published clinical study documenting improvement in ‘Parvovirus B19–Associated Chronic Fatigue Syndrome’ [55]. It is notable that the authors’ reply sets out conditions under which an effective case might be made for prescription of IgG to an ME/CFS patient on the NHS (see Box 2 below)

Box 2UK NHS IgG treatment: Process of ‘exceptionality’.*“There is a process of ‘exceptionality’ that offers the opportunity for treatment in cases that fall outside the broad definition of a disease state. In the case of parvovirus B19-associated CFS, the treating physician may request immunoglobulin treatment on the basis that parvovirus B19-associated CFS is different from the reference population of CFS patients. The physician would be expected to present evidence to the local immunoglobulin panel that an individual patients is likely to gain significantly more benefit from the intervention than might be expected for an average patient with that condition”* [80].

#### 6.1.2. United States

In the US, IgG is not Food and Drug Administration (FDA) approved for ME/CFS [83]. However, data reported in a 2006 CDC review demonstrate that much of the IgG used in the US is prescribed off label [84]. A lack of RCTs does not necessarily present a significant barrier to prescribing, as the review advises: *“Reports concerning IVIG continue to grow at a tremendous pace but few high-quality randomized controlled trials have been reported”*. It is further reported that IV IgG is most commonly used in the treatment of chronic neuropathy. Given that autoimmune-associated small-fibre polyneuropathy (aaSFPN) has been identified in some ME/CFS patients [28,30], this does appear to open a door to off label prescribing of IgG for selected ME/CFS patients.

#### 6.1.3. Australia

In Australia, where the majority of the research on use of IV Ig in ME/CFS has been conducted, access is governed by the National Blood Authority (NBA) which first produced clinical criteria for use of IgG in 2007. The current version was launched in 2018 [85]. These are broadly similar in approach to the UK criteria, with three categories of use—‘established’, ‘emerging’, and ‘exceptional’—and a further category ‘not funded’.

The position in respect of use for ME/CFS has remained the same throughout, with IgG ‘not funded’. Regarding this decision, the NBA advises: *“Ig therapy is not supported for myalgic encephalomyelitis. Results from a single RCT in 1990, have not been reproduced, and subsequent studies have shown no evidence of benefit.”* This assessment ignores the positive findings emerging from the 1997 RCT involving children and young people [22,23] (Study 3: see Section 3.3 above).

The NBA recognise that *“For a number of reasons, medical specialists may sometimes want to prescribe immunoglobulin for medical conditions that are not funded under the national blood supply arrangements”* and advise that, for such patients, Approved Health Providers may purchase imported IgG directly from the supplier at an equivalent price to that negotiated by the NBA. This requires a Jurisdictional Direct Order (see Box 3).

Box 3Australian Jurisdictional Direct Order (JDO) arrangements.Medical specialist seeks approval through a JDO for their patient, following their local processesOnce approval is granted, the relevant Approved Health Provider (AHP) places an order for the imported IVIg or SCIg directly with the supplier, establishing a contract directly between the supplier and the AHP for the supply of the productPurchases are paid for in full by the AHP.SOURCE: https://www.blood.gov.au/IgOtherAccessArrangements (accessed on 28 June 2021).

### 6.2. The Present Approach: Ongoing “General Management Strategies for Chronic Illness”

Cost has also been cited as a reason not to prescribe IgG to ME/CFS patients [18]. However, in a disease causing prolonged and substantial morbidity in many patients [54,59,60,61,62,63,64,65,66,67,68], any treatment with curative potential is likely to prove cost-effective in the long run.

Additionally the alternative direction taken by healthcare systems is not devoid of expense in terms of professional salaries. For example, in Australia—where three of the four Studies of IV IgG were conducted—Dr K. Rowe (author of Study 3) reports as follows regarding the direction subsequently taken by her paediatric service: *“As immunoglobulin was a scarce resource requiring approval by a government agency, a decision was made to not allow intravenous immunoglobulin to be available for ME/CFS for young people due to some adverse effects (Study 3), as well as inconclusive trials in adults (Studies 1,2 and 4). Thus, options for treatment reverted to general management strategies for chronic illness. …. The service has since expanded to several pediatricians and access to a 4-week self-management program run by the Victorian Pediatric Rehabilitation Service”* [86].

## 7. Conclusions

From the evidence outlined in this review, we hold that immunoglobulin is a potentially curative treatment for a proportion of patients with ME/CFS and that the interpretation of some of the literature regarding the issue has been faulty. Considering the lack of any other curative treatment for this debilitating disease, we find this regrettable.

Research on IgG for ME/CFS patients has stagnated. A severe global supply shortage of IgG has not helped; however, this is set to ease [33]. While treatment with IgG is not without cost, this has to be set against the cost of providing an ongoing healthcare service to patients with a chronic illness. An illness which, for some patients, may well cease to be chronic if treated with immunotherapy.

In 1991, a British Medical Journal editorial on use of intravenous immunglobulins projected that *“Exciting developments in treatment with intravenous immunglobulin should take place in the 1990s”* [87], referring to various disorders in this context, including myalgic encephalomyelitis. In response, a letter from Dr Charles Shepherd, medical adviser at the ME Association, highlighted two pertinent areas for investigation: *“it would seem sensible to discover whether there is a specific subgroup of patients who benefit and whether this is related to any characteristic defects in immune function”* [88].

This review has demonstrated that the original research trials provide pointers in exactly that direction, indicating that a significant proportion of patients improve and that predictors of response lie in indicators of abnormal cell-mediated immunity. While there is a degree of variation between the several research trial reports (summarised at Section 3.5), nothing has emerged in the three decades since Study 1 [13] reported recovery in 43% of IgG recipients that would substantially undermine these two core conclusions.

The phrase *“a significant proportion of patients”* is crucial. Study 1 findings do not suggest that the generality of ME/CFS patients benefit. However, for those who did, the improvements experienced were considerable (see Box 1). At the same time, because the remaining participants experienced scarcely any change, comparison of *average* outcomes recorded at final end point did not reveal a statistically significant difference between the IgG and placebo control groups, masking this key finding.

Clinical observations support this interpretation. Beneficial reports from clinical practice, evident in 1990 [18], continue to this day (Section 4). The findings emerging from our analysis represent a clear demonstration of what might be gained by approaching ME/CFS patients on the basis of documented biomedical abnormalities.

It is often asserted that a disorder exists which we call ‘CFS’ and which is heterogeneous. In our view, it would be more accurate to say that there exist several research definitions for ‘chronic fatigue syndrome’ and that these definitions, to a greater or lesser degree, have the potential to encompass a heterogeneous group. It follows that for progress to be made in identifying and treating these patients, they require investigation and differential treatment as indicated.

Parallels with ME/CFS in terms of both clinical presentation and biomarkers of immune dysfunction and neuroinflammation are emerging from the study of patients experiencing post-acute sequelae of COVID-19 (PASC) [34,35,89,90,91,92]. We therefore further conclude that it is plausible that a subgroup of PASC patients may also be helped by IgG.

## 8. Action Points Emerging

Based on the findings of this review, we suggest action on three fronts.

### 8.1. Research

Further randomised controlled trials on IgG in ME/CFS should be conducted with urgency.

Given the emergence of similarities in patients experiencing post-acute sequelae of COVID-19 (PASC), further research in this area may prove fruitful to the understanding and care of these patients also. Researchers may wish to incorporate a comparison of associated groups in RCT design. For example:ME/CFS patients and patients experiencing PASC.ME/CFS patients who have gone on to suffer from COVID-19 and ME/CFS patients who have not.ME/CFS patients who have gone on to suffer from COVID-19, ME/CFS patients who have not and patients experiencing PASC.

### 8.2. Laboratory Testing

Healthcare systems should encourage the identification of ME/CFS patients with abnormal cell-mediated immunity (CMI) and/or an autoimmune-associated small-fibre polyneuropathy (aaSFPN); this may facilitate identification of likely responders to IgG.

The measures of CMI employed to assess subjects in the IgG trials are documented in Appendix B. While a full consideration of present day testing regimes is beyond the scope of this paper, further tests—such as autonomic receptor auto-antibody assessment—may be pertinent.

### 8.3. Treatment

Pending further research and introduction of laboratory testing, it would be reasonable to offer a therapeutic trial of IgG to selected ME/CFS patients at the more severe end of the spectrum based on clinical presentation.

## Figures and Tables

**Table 1 healthcare-09-01546-t001:** IV immunoglobulin trial characteristics.

Trial Characteristic	Study 1 Lloyd et al., 1990 [13]	Study 2 Peterson et al., 1990 [14]	Study 3 Rowe 1997 [22]	Study 4 Vollmer-Conna et al., 1997 [24]
Immunoglobulin G (IgG) dosage	2 g per kg	1 g per kg	1 g per kg, to a maximum of 60 g	2 g, 1 g, or 0.5 g per kg
IgG Brand	Intragram™	Gammagard^®^	Intragram™	Intragram™
Placebo infusion	unclear	1% albumin solution	1% albumin in a 10% weight/volume maltose solution ^1^	1% albumin in a 10% weight/volume maltose solution ^1^
Number and frequency of infusions	3 at monthly intervals	6 at 30 day intervals	3 at monthly intervals	3 at monthly intervals
Participants recruited	*n* = 49 23 IgG 26 Placebo	*n* = 30 15 IgG 15 placebo	*n* = 71 36 IgG 35 Placebo	*n* = 99 23 at 2 g IgG per kg 28 at 1 g IgG per kg 22 at 0.5 g IgG per kg 26 placebo
Drop outs	2 from the IgG group	1 from each group	none 1 lost to follow-up	3 IgG recipients (dosage not specified)
Findings based on	*n* = 49 ‘intention to treat’	*n* = 28	*n* = 70	*n* = 99 ‘intention to treat’
Timing of final follow-up	3 months after final infusion + ‘responders’ reviewed at 6 and 12 months	within 48 h of final infusion	6 months after final infusion ^1^	3 months after final infusion

^1^ The IgG brand used, Intragram™, is likewise prepared in a 10% weight/volume maltose solution. A report including feedback on participant experiences over the five years following participation was published in 1999 [23]; however, it was not possible to carry out a placebo vs. intervention group comparison at this time as many placebo group patients were subsequently treated with IgG.

**Table 2 healthcare-09-01546-t002:** IV immunoglobulin trial participant characteristics.

Participant Characteristic	Study 1 Lloyd et al., 1990 [13]	Study 2 Peterson et al., 1990 [14]	Study 3 Rowe 1997 [22]	Study 4 Vollmer-Conna et al., 1997 [24]
Age range	16–63	Eldest 74, youngest not stated	11–18	16–73
Duration of illness	Range 1–15 years median 47 months	Mean 3.8 years	mean in months: IgG 19.2 placebo 16.9	mean in years: IgG 2 g per kg 5 IgG 1 g per kg 7 IgG 0.5 g per kg 6 placebo 7
Illness onset	Acute viral-like onset in 37 (76%)	All but one began with episode of an acute viral-like illness	Over 85% developed the illness secondary to an infection	Acute viral-like onset in 75 (76%)
Severity at baseline	*“well-characterised, severe, and long-standing CFS”*	Not directly reported, but apparently less severe that was the norm for clinic (see text for details)	*“by selection at the more severe end of the spectrum”*	77% able to work 28% working more than half time; average 5 h of non-sedentary activity daily
Recruitment criteria	Australian 1988 [37] (Lloyd et al.), in which marked exercise-induced muscle fatigue with prolonged recovery time is essential	Centers for Disease Control (CDC) 1988 [36] (Holmes et al.)	CDC 1994 [38] (Fukada et al.) Also: identifiable time of onset; exacerbated by minor exercise	Lloyd et al., 1990 [37] and Schluderberg et al., 1992 [38] ^1^
Cohort illness characteristics	All chronic and persisting; none relapsing and remitting	All but one reported prolonged post-exertional fatigue In addition to fatigue, had been experiencing an average of 8.8 of the CFS symptoms listed by the CDC [36]		*“probably, more heterogeneous than that enrolled in our earlier study”*

^1^ An account of the National Institutes of Health workshop which initiated the development of the CDC criteria, subsequently published in 1994 [39].

**Table 3 healthcare-09-01546-t003:** Incidence of clinically significant improvement, Rowe 1997 [22].

Period after Final Infusion	Gammaglobulin Group (*n* = 36) ^1^		Placebo Control Group (*n* = 35) ^2^	
	Not Improved	Improved	Not Improved	Improved
3 months	17 (47.2%)	19 (52.8%)	24 (68.6%)	11 (31.4%)
6 months	10 (27.8%)	26 (72.2%)	19 (55.9%)	15 (44.1%)

^1^ The therapeutic agent is referred to as ‘gammaglobulin’ in this paper; this was the same IgG preparation as used in Studies 1 and 4. ^2^
*n* = 34 at six months as one participant was lost to follow-up after the three month assessment.

## Data Availability

Not applicable.

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
