# Peer review of "Back to the Future? Immunoglobulin Therapy for Myalgic Encephalomyelitis/Chronic Fatigue Syndrome"

_healthcare, 2021, doi:10.3390/healthcare9111546_

Round 1
Reviewer 1 Report
Overall, this report is reasonable in its proposal, supportive evidence and conclusions and the subject matter does need to be debated. I wonder whether modern therapies such as those employing aptamers and monoclonal antibodies etc may soon obviate the need for using immunoglobulin therapy (IgT) as these are much less expensive to produce and can target specific molecules, auto-antibodies etc. The role of IgT in CFS/ME should have been debated 20 years ago but is still worthy of discussion until we have better targeted therapy for CFS/ME.
I agree that the previous trials on immunoglobulin therapy (IgT) in CFS/ME would now be considered inadequate in terms of proper patient selection, measures of assessing fatigue and duration of therapy and that a proper trial is needed. I also agree that the conclusions formed by the medical/scientific community on using IgT in CFS/ME based on these studies is incorrect.
As we do not presently understand the precise nature of the immune dysfunction in CFS/ME (and this may be different between patients and indeed over the course of the illness), it is unclear how immunoglobulin therapy works. With impaired NK cell function being the most consistently reported abnormality, the immunoglobulin therapy may improve antibody dependent cytotoxicity against viral or other microbial pathogens where this is inadequate in the patient. Re-regulation of the immune system, anti-inflammatory actions, reduction in auto-antibody levels and formation would represent some of the other relevant possibilities. I also agree that the dangers of IgT have been overplayed with a view to understandably limiting the use of this scarce resource for those with definite and obvious antibody deficiency/systemic autoimmunity. This has made it virtually impossible to use IgT in several conditions including CFS, where immune dysfunction has not been definitively demonstrated. However, in my experience only a very small proportions of patients have had reactions and these have been relatively mild and confined to those with symptomatically active lung infection and an underlying antibody deficiency. This is not the case for CFS. Thus, denying its use on this basis is unreasonable. The same is true for the risk of long term infections of unknown viruses/prions etc which has not become evident since pre and post production tests of known pathogens has been undertaken and production methods have including a solvent extraction, pasteurisation stage and nanofiltration etc. In the absence of any other treatment providing significant benefit, I think the value of IgT in CFS/ME should be properly re-investigated soon.
The focus of the present submitted report has been on the immune aspects of IgT. However, it would be important to recognise that the benefit of the IV IgT, and to a lesser degree of the IM IgT, may also be due to mechanisms other than those relating to immune function. Thus, IVIg has an oncotic pressure similar to plasma and thus has the ability to expand the intra vascular blood volume for a significant period of time. Importantly, the intravascular volume is contracted in people with severe CFS/ME as a result of endocrine and cardiovascular changes and especially in those with an underlying joint hypermobility syndrome. IgT may therefore improve the perfusion of vital organs such as the brain and would work similar to regular normal saline infusions which can also help many people with CFS/ME. I would suggest that the precise ingredients of the placebo are also recorded in the table of the different studies as those with high oncotic pressure eg albumin may act in a similar way to the IVIg. Therefore, the differences between the immunoglobulin and placebo groups may be reduced and with both groups improving as in study 4.
I would ask for the following changes:
- Detail what precise tests of cell mediated immunity were used in the different reports.
- I would also ask that the authors suggest that more up to date tests of immune dysfunction eg NK cell enumeration/cytotoxicity testing, autonomic receptor auto-antibody assessment etc be employed in future studies of IgT in CFS/ME.
- As indicated above I would suggest that the constituents of the placebo are mentioned and for study 4 in particular how was blinding (if any) conducted?
Reviewer 2 Report
This paper presents a review of research trials for the disorder called "chronic fatigue syndrome (CFS)" that were published between 1990-1999.
The 4 case studies introduced in the paper are described comprehensively with sufficient details. Based on the evidence outlined, it seems that some early definitions of CFS are heterogeneous under different studies, and the interpretation of some of the literature regarding the issue has been faulty.
The review provides a key insight: as a subgroup of post-acute sequelae of COVID-19 (PASC) patients may be helped by IgG, deeper understanding on IgG is needed with urgency, and thus RCTs on IgG in ME/CFS should be conducted.
Overall, the motivation of the paper is clear and it is well presented with detailed analysis and discussion.
Reviewer 3 Report
Authors may additionally consider suggesting research designs that include:
(1) ME/CFS patients that have contracted COVID-19;
(2)ME/CFS patients and COVID-19 (Long-Haul) patients.
(3) ME/CFS patients, ME/CFS patients that have contracted COVID-19 and COVID-19 (Long-Haul) patients.
